



# Nutrient utilization and diatom productivity changes in the low-latitude SE Atlantic over the past 70 kyr: Response to Southern Ocean leakage

Katharine Hendry[1], Oscar Romero[2,3], and Vanessa Pashley[4]

[1]University of Bristol, School of Earth Sciences, Wills Memorial Building, Queen's Road, Bristol, BS8 1RJ, UK
[2]MARUM–Center for Marine Environmental Sciences, Leobener Str. 8, University of Bremen, 28359 Bremen, Germany
[3]Alfred Wegener Institute, Helmholtz Centre for Polar and Marine Research, 27568 Bremerhaven, Germany
[4]Geochronology and Tracers Facility, British Geological Survey, Keyworth, NG12 5GG, UK

**Correspondence:** Katharine Hendry (K.Hendry@bristol.ac.uk)

**Abstract.** Eastern Boundary Upwellings (EBUs) are some of the key loci of biogenic silica (opal) burial in the modern ocean, representing important productive coastal systems that extraordinarily contribute to marine organic carbon fixation. The Benguela Upwelling System (BUS), in the low-latitude SE Atlantic, is one of the major EBUs, which is under the direct influence of nutrient-rich Southern Ocean waters. Quantification of past changes in diatom productivity through time, in response to Late Quaternary climatic change, feeds into our understanding of the sensitivity of EBUs to future climatic perturbations. Existing sediment archives of silica cycling include: opal burial fluxes, diatom assemblages and opaline silicon isotopic variations (denoted by $\delta^{30}$Si). Burial fluxes and siliceous assemblages are limited to recording the remains reaching the sediment (i.e. export), and $\delta^{30}$Si variations are complicated by species-specific influences and seasonality. Here, we present the first species-specific $\delta^{30}$Si record from the BUS, encompassing full glacial conditions to the Holocene. In addition to export, our new data allows us to reconstruct utilisation of dissolved Si in surface waters in an area with strong input from Southern Ocean waters. Our new archives show that there was enhanced upwelling of Southern Ocean Si-rich water, and accompanied strong silicic acid utilisation by coastal dwelling diatoms, during Marine Isotope Stage 3 (60-40 kyr). This pulse of strong silicic acid utilisation was followed by a weakening of upwelling and coastal diatom Si utilisation into MIS2, before an increase in pelagic diatom Si utilisation across the deglaciation. We combine our findings with mass balance model experiments to show that changes in surface water silica cycling through time are a function of both upwelling intensity and utilisation changes, illustrating the sensitivity of EBUs to climatic change on glacial-interglacial scales.

## 1 Introduction

Marine productivity by diatoms represents up to half of the total fixation of organic matter in the oceans and plays a key role in uptake of carbon dioxide ($CO_2$) from the atmosphere (Tréguer et al., 2018). Large-scale oceanic circulation is a first-order control on the present-day supply of dissolved silicon (silicic acid or DSi) to surface waters that is essential for diatom growth. In the modern ocean, the burial of biogenic silica (opal) is localized in regions characterised by a supply of DSi-rich waters: the Southern Ocean (SO), subarctic Pacific, and in Eastern Boundary Upwellings, EBUs (Ragueneau et al., 2000). The



Benguela Upwelling System (BUS; Fig. 1) in the SE Atlantic is one of the major EBUs, and is the under the influence of DSi-rich SO waters (Berger and Wefer, 1996; Berger et al., 2002). Quantification of BUS Si production in the Late Quaternary

is important for understanding the functioning of this highly-productive EBU, and the sensitivity of the strong and dynamic biological production to changes in wind-driven mixing and SO leakage and ventilation (Hendry and Brzezinski, 2014). The silicic acid leakage hypothesis (SALH) postulates that shifts in the leakage (i.e. export) of DSi, relative to other nutrients, within Antarctic Intermediate Water (AAIW; Fig. 1) could occur on glacial-interglacial timescales as a result of changes in SO diatom physiology and silicification due to alleviation of iron stress from dust supply (Brzezinski et al., 2002). The northward

supply of waters with a higher DSi-to-nitrate ratio via AAIW during full glacials would promote low latitude diatom growth at the expense of other non-siliceous phytoplankton, resulting in a weakening of the carbonate pump, changing seawater alkalinity and contributing to atmospheric $pCO_2$ drawdown (Matsumoto and Sarmiento, 2008). Opal burial and geochemical archives show that DSi leakage during the last glacial maximum (LGM) is variable, with some evidence for this mechanism in the South Pacific (Chase et al., 2003; Rousseau et al., 2016), but patchy opal burial response in the Equatorial Pacific

(Kienast et al., 2006). In the Atlantic, there is stronger evidence that DSi leakage and opal production was higher during the deglacial rather than the LGM (Hendry et al., 2016; Meckler et al., 2013). The impact of DSi leakage on opal burial during the deglacial is heterogeneous, both in the Atlantic and Pacific (Bradtmiller et al., 2006, 2007; Dubois et al., 2010), suggesting that ventilation of DSi-rich waters is required to promote diatom growth (Hendry and Brzezinski, 2014). Fewer archives exist for Late Quaternary DSi leakage under full glacial conditions, although one record highlights potential SO leakage and increase

in AAIW DSi in Marine Isotope Stages (MIS) 3/4 in the tropical NW Atlantic (Griffiths et al., 2013). Analogous to carbon cycling, siliceous production can be considered as two interconnected processes: net utilisation is the proportion of DSi taken up by diatoms to form opal in ocean surface waters, and export production is the opal that sinks out of the surface waters into the deep (Ragueneau et al., 2000). However, when reconstructing marine siliceous primary production, the only evidence available is the opal buried as siliceous remains of microorganisms within marine sediments. Understanding the link between surface

DSi utilisation, recycling, and export of opal is important to understand ecosystem function and organic matter sequestration. The stable Si isotopic composition (denoted in per mil relative to standard NBS28/RM8546 by $\delta^{29}Si$ or more commonly $\delta^{30}Si$, Equ. 1, 2) of opal provides an archive of DSi utilisation, given the preferential uptake of lighter Si isotopes during diatom biomineralisation (De La Rocha et al., 1997).

$$\delta^{29}Si(\text{‰}) = \left( \frac{\left(\frac{^{29}Si}{^{28}Si}\right)_{Sample}}{\left(\frac{^{29}Si}{^{28}Si}\right)_{NBS28}} - 1 \right) \times 1000 \tag{1}$$





$$\delta^{30}\text{Si}(\text{‰}) = \left( \frac{\left( \frac{^{30}\text{Si}}{^{28}\text{Si}} \right)_{\text{Sample}}}{\left( \frac{^{30}\text{Si}}{^{28}\text{Si}} \right)_{\text{NBS28}}} - 1 \right) \times 1000 \tag{2}$$

Culture studies and field observations indicate that diatoms discriminate Si isotopes by a fractionation factor, e, of approximately -1.1 ‰, although estimates for this value range from -0.4 to -2.5 ‰ (Hendry and Brzezinski, 2014, & references therein). The application of this utilisation proxy is complicated by several unknowns: changes in the initial isotopic composition of ambient seawater through time (Horn et al., 2011); potential seasonal and ecological bias (Swann et al., 2017); and potential species-specific fractionation of Si isotopes in mixed assemblages (Sutton et al., 2013). Here, we present the first late Quaternary $\delta^{30}$Si records from the BUS during the full glacial conditions of Marine Isotope Stage (MIS) 4 to the late Holocene, overcoming these challenges by using only two large centric species from a well-documented sediment core. We will use these records, together with a simple box model, to reconstruct past changes in seawater composition and DSi supply through time.

## 2 Material and Methods

Gravity core GeoB3606-1 was collected in the SE Atlantic Ocean (BUS, Fig. 1, 25°28′S, 13°05′E, 1785 m water depth), providing an exceptionally high-resolution paleoclimatic archive of nutrient conditions, productivity, and sea surface temperature (SST) variations off Namibia for the past 70 kyr (McKay et al., 2016; Romero et al., 2015, 2003; Romero, 2010; Shukla and Romero, 2018, Supplemental Material). The robust age model for core GeoB3606-1 has been published elsewhere (Romero, 2010; Romero et al., 2015), with additional radiocarbon dates from McKay et al. (2016). The published conventional [14]C ages have been converted to calendar ages. High-resolution counts of diatom valves, bulk biogenic components, planktonic and benthic foraminifera stable isotopes and alkenone-based SST have been previously published (Romero, 2010; Romero et al., 2015; McKay et al., 2016). At site GeoB3606-1, diatoms are the main contributors to the opal fraction. The highest diatom accumulation rate and exceptionally high values of opal (up to 45 wt.%), as well as the strongest millennial and sub-millennial scale wt.% fluctuations, occurred from 70 and 30 kyr (Fig. 2). Changes in SST and mixing of the uppermost water column alone are unlikely to fully explain variations in diatom productivity (Romero et al., 2015). The inverse correlation between the relative abundance of the Antarctic diatom *Fragilariopsis kerguelensis* and the alkenone-based SST variations (Fig. 2) in GeoB3606-1 from 70 to 30 kyr suggests enhanced invasion of DSi-rich SO waters and stronger wind-driven mixing during this interval of high opal burial (Romero et al., 2003; Shukla and Romero, 2018). The warming of the last deglaciation (19-13 kyr, Fig. 2) suggests, in contrast, stratification of the uppermost water column, accompanied by a distinctive shift in the diatom assemblage from an upwelling-dominated to a non-upwelling community (Fig. 2), and an increase of calcareous production (Romero et al., 2003). To better understand these past changes, and deconvolve interactions between DSi supply, utilisation in





surface waters and burial in sediments, we constructed a record of $\delta^{30}$Si from hand-picked specimens of *Actinocyclus curvat-*
*ulus* and *Coscinodiscus radiatus* at site GeoB3606-1 (Fig. 2). Single specimen (or mono-generic) $\delta^{30}$Si diatom archives have
previously been constructed to assess Si utilisation in particular environments, and reliably record different absolute values and
trends compared to measurements from bulk or other siliceous fractions (Doering et al., 2016a, b; Hendry et al., 2014; Xiong
et al., 2015). *A. curvatulus* is a coastal diatom, mainly thriving in shallow, hemipelagial waters bathing the uppermost slope off
Namibia. According to multiyear sediment trap studies off Mauritania, *A. curvatulus* has a higher contribution to the diatom
community before and after the main upwelling season (Romero and Fischer, 2017). *C. radiatus* is a common component
of planktonic assemblages of tropical and subtropical hemi- to pelagial marine areas (Romero and Fischer, 2017). Compared
to *A. curvatulus*, it represents a more "pelagial" signal. Both diatoms are "petri dish-shaped" and can be considered "large":
diameter up to 200-220 $\mu$m (Hasle et al., 1996). Their valves are significantly larger than that of upwelling diatoms (resting
spores of *Chaetoceros*, up to 25-30 $\mu$m length) and other coastal planktonic taxa (25-50 $\mu$m). This also means that one *A. cur-*
*vatulus* or *C. radiatus* valve contains possibly 10 to 20 times more opal than the delicate spores of *Chaetoceros* and consume
relatively large amounts of DSi in the build-up of their valves compared to coastal upwelling diatoms. As both species grow
away from the major upwelling zone, their isotopic compositions (denoted by $\delta^{30}$Si$_{CA}$ for *Coscinodiscus/Actinocyclus*) will
reflect a combination of the initial composition of the ambient water, and their DSi utilisation. Utilisation by both species is
unlikely to have a quantitative impact on ambient seawater composition, due to very low overall abundances throughout the
record (Romero et al., 2003). Instead, the $\delta^{30}$Si$_{CA}$ will reflect utilisation by the dominant upwelling species (resting spores of
*Chaetoceros* spp.).

## 2.1   Laboratory methods

*A. curvatulus* and *C. radiatus* valves were hand-picked from washed and sieved sediments (note that low MIS2 data resolution
was due to limited sample availability). The picked diatoms were transferred to cleaned Teflon vials and any organic material
was removed by drying down in concentrated nitric acid (in-house twice-distilled $HNO_3$) on a hotplate. The diatoms were
dissolved in a strong alkaline solution (0.4M sodium hydroxide Analar) at 100°C overnight. The resulting solution was diluted
two-fold in in 18 M $\Omega$.cm Milli-Q water and acidified using 6N hydrochloric acid (in-house twice-distilled HCl) to reach a pH
of 2-3. The samples were then purified using cation exchange resin (Georg et al., 2006). Analysis of silicon and magnesium
isotopes ($^{28}$Si, $^{29}$Si, $^{30}$Si, $^{24}$Mg, $^{25}$Mg, and $^{26}$Mg) was carried out in a pilot study by Multi-Collector Inductively-Coupled
Plasma Mass Spectrometry (Cassarino et al., 2018; Hendry et al., 2015) at the University of Bristol (Thermo Neptune). Further
analyses were carried out using similar methodology at the NERC Isotope Geosciences Laboratory. Each sample was filtered,
prior to isotopic analysis, to remove any fine particles, which may have eluted off the cation exchange column during the
purification stage (Millex-LG, 0.2 $\mu$m, PTFE syringe filter, Millipore). Samples and reference materials were acidified using
HCl (to a concentration of 0.05M, using twice quartz distilled acid) and sulphuric acid (to a concentration of 0.003M, using
Romil Ultra Purity Acid). This was done following the recommendations of Hughes et al. (2011), the principle being that
swamping both samples and reference materials, above and beyond the natural abundance of Cl$^-$ and SO$_4{}^{2-}$, will evoke a
similar mass bias response in each. Finally, all samples are doped with 300ppb magnesium (Mg, Alfa Aesar SpectraPure).





Spiking with an external element of known isotopic composition allows the data to be monitored and corrected for the effects of instrument induced mass bias (Cardinal et al., 2003). In order to resolve isobaric interferences, principally $^{14}N^{16}O^+$ at

mass 30, samples were analysed using the medium mass-resolution capability of a Thermo Scientific Neptune Plus MC-ICP-MS (multi collector inductively coupled plasma mass spectrometer), operated in wet-plasma mode. Instrument and sampling details are summarised in Table 1. Using the instrumental parameters outlined, a sensitivity of approximately 4.5V/ppm was obtained. Data were acquired using a dynamic, two sequence acquisition (see Tables 1 and 2 for full operating conditions). Faraday amplifier gains were measured at the beginning of each analytical session. Data were collected in 1 block of 25 ratios,

with a resulting analysis time of approximately 12 minutes per sample (including the sample uptake and stabilisation time of 90 seconds). Blanks were measured on the sample make-up acid (0.05M HCl, 0.003M $H_2SO_4$) using a shortened version of the acquisition procedure above (1 block of 10 ratios). An on-line background correction was made, with the values obtained for the blank acid subtracted from each succeeding sample. Isotope ratios were calculated using following Equ. 1 and 2. Reference standards were run to assess the accuracy and precision of the technique (NIGL diatomite $\delta^{30}$Si +1.20 ± 0.16 ‰ (2SD, n = 12);

Bristol diatomite +1.30 ± 0.07 ‰ (2SD internal error, n = 1); Bristol LMG08 sponge standard -3.43 ± 0.08 ‰ (2SD internal error, n = 1)), in good agreement with published values (Reynolds et al., 2007; Hendry et al., 2011). Complete replicate measurements were carried out from three different horizons, and reproduced within 0.04 to 0.39 ‰, indicating the natural level of isotopic heterogeneity within a sediment sample. The $\delta^{29}$Si and $\delta^{30}$Si values of standards and samples showed mass dependent behaviour (slope of 3-isotope plot of 0.50 ± 0.06, consistent with equilibrium or kinetic fractionation; Reynolds

et al. (2007)). The procedural blank was below the level of detection.

## 2.2  Modelling

A two-box model "thought experiment" was devised in MATLAB to investigate changes in upwelling and biological productivity between simulated MIS3 and late MIS3/MIS2 conditions, based on De La Rocha and Bickle (2005). The model comprised a surface box (area 100km x 100km, with variable depth), and a deep box (water column height 2500m, area 100km x 100km).

The conditions are described in Table 3.

A "MIS3" spin up is run for 50 thousand years to reach steady-state, with a deep surface box (100m), high biological production efficiency (i.e. proportion of available DSi used by diatoms) was set at 90%, strong SO input (approximately 1.5 x modern) and export efficiency of 50% (similar to modern). DSi input is set at 70 $\mu$M with a $\delta^{30}$DSi of +1.2 ‰, reflecting low utilisation in the SO during periods of silicon leakage. Note that although culture studies of *Chaetoceros brevis* revealed a

fractionation factor of -2.09 ‰ during growth (Sutton et al., 2013), the fractionation during *Chaetoceros* resting spore formation is unknown so the bulk diatom fractionation is set at -1.1 ‰. Conditions are then changed to simulate the transition into late MIS3/MIS2 ('MIS2'): SO input is dropped to modern concentrations and isotopic composition and biological production efficiency is decreased, and the volume of the surface box is decreased. To test the sensitivity of the system systematically, each of the following parameters was altered one at a time: utilisation of diatoms, proportion of SO input, [DSi] of SO input,

upwelling rate, and volume of the surface box.



## 3 Results and Discussion

The downcore $\delta^{30}\text{Si}_{CA}$ values range from -1.5 to +1.5 ‰, with the lightest isotopic compositions after 39 kyr and the heaviest between 40-50 kyr and after 12 kyr (Fig. 2; supplementary data table). The model output values after 2000 years are shown in Fig. 3. These experiments reveal that $\delta^{30}\text{Si}_{CA}$ was somewhat sensitive to changes in SO input of [DSi], and upwelling rate,

with reasonable variation within the model of these parameters allowing for a change in $\delta^{30}\text{Si}_{CA}$ of approximately 0.4 ‰ in each case. Changing the surface box volume could achieve a change in $\delta^{30}\text{Si}_{CA}$ of approximately 0.8 ‰. Values of $\delta^{30}\text{Si}_{CA}$ were most sensitive to utilisation (a drop from near complete utilisation to 10% utilisation resulted in a decrease in $\delta^{30}\text{Si}_{CA}$ of 1.1 ‰). However, the very isotopically light compositions (where $\delta^{30}\text{Si}_{CA} < 0$ ‰) observed in the downcore archive were not achieved by altering only a single parameter at a fixed diatom fractionation factor.

### 155 3.1 Diatom utilisation intervals in the BUS for the past 70 kyr

Using a combination of the new isotopic records, together with published values for opal accumulation and diatom assemblage composition, we can recognise four main intervals within our $\delta^{30}\text{Si}_{CA}$ DSi utilisation archive (Fig. 4).

#### 3.1.1 Marine Isotope Stage (MIS) 4 (70-60 kyr)

During MIS 4, $\delta^{30}\text{Si}_{CA}$ values are relatively high (approximately +1.5 ‰) but exhibit a pronounced excursion from 70-63 kyr

towards lighter isotopic compositions, reaching a minimum of approximately 0 ‰ at 66 kyr. SST records indicate a pulse of upwelling, accompanied by an increase in diatom accumulation, coincident with this excursion. This suggests that there was a transient period of relatively low utilisation by the dominant small-sized *Chaetoceros* spp. as upwelling and export of opal intensified into the early MIS3

#### 3.1.2 Early MIS 3 (60-40 kyr)

The early MIS3 is characterised by high but variable $\delta^{30}\text{Si}_{CA}$, ranging between +0.5 and +1.5 ‰. These high $\delta^{30}\text{Si}_{CA}$ values are accompanied by an increase in *F. kerguelensis* and low SST, indicative of strong utilisation together with a greater input of DSi-rich SO water and intense upwelling. Strong *Chaetoceros* utilisation would have reduced the concentration of pre-formed DSi exported away from the shelf and uppermost slope waters towards the *A. curvatulus* and *C. radiatus* habitats (hemipelagial and pelagial). Not only was *Chaetoceros* production high because of the rate of supply of DSi to coastal waters,

but also because they were able to use a high proportion of what was available. High opal burial is also found during this time interval in sediments from the Eastern Equatorial Pacific (Kienast et al., 2006), perhaps indicating that this mechanism could have been active in other upwelling zones in the open ocean during MIS3. A similar mechanism may also have been active in the BUS during the Plio-Pleistocene (Etourneau et al., 2012). The time lag (approximately 10 kyr) between the decline in diatom accumulation and $\delta^{30}\text{Si}_{CA}$ could be due to a decrease in the DSi concentration of the supplied water after 49 kyr (i.e.

a decline in SO water, indicated by the decrease of *F. kerguelensis* abundance, Fig. 2). High utilisation of waters with a lower DSi concentration would result in lower total opal production and would also potentially contribute towards higher $\delta^{30}\text{Si}_{CA}$





values due to prior Si isotopic enrichment of the water. The resolution of $\delta^{30}\mathrm{Si}_{CA}$ samples also declines into MIS 3 and MIS2, as a result of low diatom abundance, which limits our ability to link temporally the diatom utilisation and opal production throughout this period.

### 180  3.1.3  Late MIS 3 to 2 (40-15 kyr)

By the late MIS3, the $\delta^{30}\mathrm{Si}_{CA}$ record starts to decline towards lower values, reaching the lowest value of -1.5 ‰ by the end of MIS2 (Fig 2). Weakening upwelling (trends to higher SST) occurred at the same time as the decline towards lower $\delta^{30}\mathrm{Si}_{CA}$ values, which points towards a significant decrease in utilisation as well as low opal production: as less DSi was being supplied from the SO, diatoms generally decline in abundance. MIS2 saw a shift from a siliceous–calcareous productivity system to one
dominated by calcareous production began during early MIS2 (Romero et al., 2003).

### 3.1.4  MIS 1 (15 kyr to present)

MIS1 is characterised by a return to higher $\delta^{30}\mathrm{Si}_{CA}$ values (approximately 0 to +1.5 ‰), similar to MIS4 and early MIS3 (Fig. 2A), suggesting a return to high DSi utilisation. The dominance of open-ocean, warm-water diatoms, coupled with low diatom and opal accumulation throughout MIS1, has been interpreted to reflect the predominance of DSi-poor water masses at 25°
in the BUS (Romero, 2010). At the same time as the decline in upwelling intensity (from SST and diatom assemblage), and overall low SO water contribution, our new isotope record shows a step change into MIS1 to higher $\delta^{30}\mathrm{Si}_{CA}$ values, similar to MIS3/4 (Fig. 2). This suggests that there was an increase in pelagic diatom utilisation, despite lessened DSi availability and weakened mixing. This is potentially also coupled with a reduction in DSi concentration and accompanying increase in initial seawater Si isotopic composition. A drop in *Chaetoceros* utilization (of a less enriched DSi water supply, supplied at a lower
rate) will result in more DSi being exported offshore from the upwelling zone and a higher DSi availability for the (hemi) pelagic diatoms, which will thrive under more favourable conditions. The high MIS1 $\delta^{30}\mathrm{Si}_{CA}$ values are likely, possibly for the first time in the record, to reflect the utilisation by pelagic diatoms, including both *A. curvatulus* and *C. radiatus*.

### 3.2  Quantifying changes to the BUS silica cycle

To quantify the potential sensitivity of the BUS to these different driving mechanisms, we have constructed a mass balance
thought-experiment, based on a two-box model. DSi enters the system from rivers into the surface box and via 'upwelling' inputs (i.e. SO water) into the deep box. The deep and shallow boxes are joined by DSi fluxes, and opal is produced by diatoms in the surface box (assuming utilisation efficiency and preservation), fractionating Si isotopes by -1.1 ‰ (De La Rocha et al., 1997). The experiments highlight that either a change in input seawater composition or a reduction in utilisation alone are unlikely to be able to drive the large downcore shifts in $\delta^{30}\mathrm{Si}_{CA}$ shifts (approximately 3 ‰), and some other additional feedback
mechanism, coupling inputs and biological uptake, is required. A combination of changes in upwelling, stratification, seawater input and utilisation can act together to change the isotopic differentiation of shallow and deep-water masses. The experiment results also indicate that, in order to achieve the extremely low $\delta^{30}\mathrm{Si}_{CA}$ values observed downcore, isotopic fractionation



during DSi uptake by *A. curvatulus* and *C. radiatus* is likely to be greater than generally assumed, up to -2 ‰, as observed in some diatom cultures of other species (Sutton et al., 2013).

### 3.3 Response of the BUS to silicic acid leakage

In addition to informing on silica cycling within the BUS ecosystem, our downcore silicon isotope archive provides broader insight into the SALH. We interpret the diatom $\delta^{30}\text{Si}_{CA}$ record as indicating that there was strong but variable upwelling of Si-rich waters during MIS4 and MIS3, consistent with leakage of SO waters at this time into the eastern basin of the South Atlantic. If correct, this interpretation would further suggest that this Si-enriched AAIW could have been exported throughout the Atlantic, given the reconstructed shifts in BUS upwelling and utilisation are coincident in time with previous downcore evidence of silica leakage into the western basin during MIS3/4 (Griffiths et al., 2013). This silica leakage did not appear to drive strong diatom production throughout the Atlantic, however, and only influenced diatom production in any significant way in the eastern basin, where the thermocline was sufficiently shallow (Abrantes, 2000; Flores et al., 2000; Abrantes, 2003). Despite this, the silica leakage could have driven an increase in diatom production relative to other phytoplankton groups in regions of the eastern Atlantic basin, contributing to the drop in atmospheric $CO_2$ observed in ice cores and the decline into full glacial conditions and so supporting the SALH (Matsumoto and Sarmiento, 2008). However, we would suggest that physical oceanographic changes are likely to be largely responsible for the decline in $CO_2$ during MIS4, which occurred over a much shorter timescale than both the changes in silica cycling within the BUS and the evidence for silica leakage in the eastern basin (Griffiths et al., 2013; Thornalley et al., 2013). Furthermore, the export of Si-enriched SO waters did not last the duration of the last glaciation, and began to decline into late MIS3/MIS2 (i.e. into the LGM) again arguing against a key driving role for biological carbon uptake in controlling atmospheric $CO_2$.

### 3.4 Implications for Paleonutrients and Productivity

The cycling of Si in surface waters, and the relationship with organic matter production, will be a function of both net production (utilisation efficiency and recycling) and export. EBUs are generally characterised by high utilisation and export, with rapid turnover. However, the intensity of upwelling and nutrient composition of upwelled waters through time are likely to be sensitive to climate forcings, and will impact the balance between utilisation and export, the availability of preformed DSi, and the coupling between Si and C cycles (Romero et al., 2003; Romero, 2010; Hendry and Brzezinski, 2014). Using a combination of $\delta^{30}\text{Si}$, diatom assemblages, opal accumulation, and alkenone-based SST allows investigation of how the diatom community was using supplied DSi in the surface waters during periods of rapid climate variations in low-latitude EBUs. We have shown that this approach can be successfully used in the BUS to produce a continuous record across full glacial conditions to investigate the impact of SO leakage and ventilation on diatom production in surface waters. This approach is a promising means to assess the impact of water mass variations on Si cycling by diatoms in other environmental settings, where downcore diatom assemblages have already been recorded. Upwelling areas (e.g. Equatorial regions and frontal zones) are ideal targets, given their sensitivity to rapid changes in ocean mixing and subsurface nutrient supply. By carefully selecting identifiable and



well-constrained diatom species for isotopic analysis, DSi utilisation and production can be quantified more robustly, without uncertainties arising from seasonality, niche conditions, or species-specific fractionation.

*Data availability.* All data are available for download at https://doi.pangaea.de/10.1594/PANGAEA.921237

*Author contributions.* OR and KH devised the study, KH and VP carried out the isotope analyses. All authors were involved in the preparation of the manuscript

*Competing interests.* The authors declare that they have no conflict of interest.

*Acknowledgements.* The authors would like to thank Stephanie Bates for help in the laboratory and Gregory de Souza for advice on box modelling; KH is supported by the Royal Society (URF/R/180021 and RG130386). OER is supported by DFG grant RO3039/9-1. Lisa Mehring is acknowledged for her work in preparing samples for Si measurements.





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





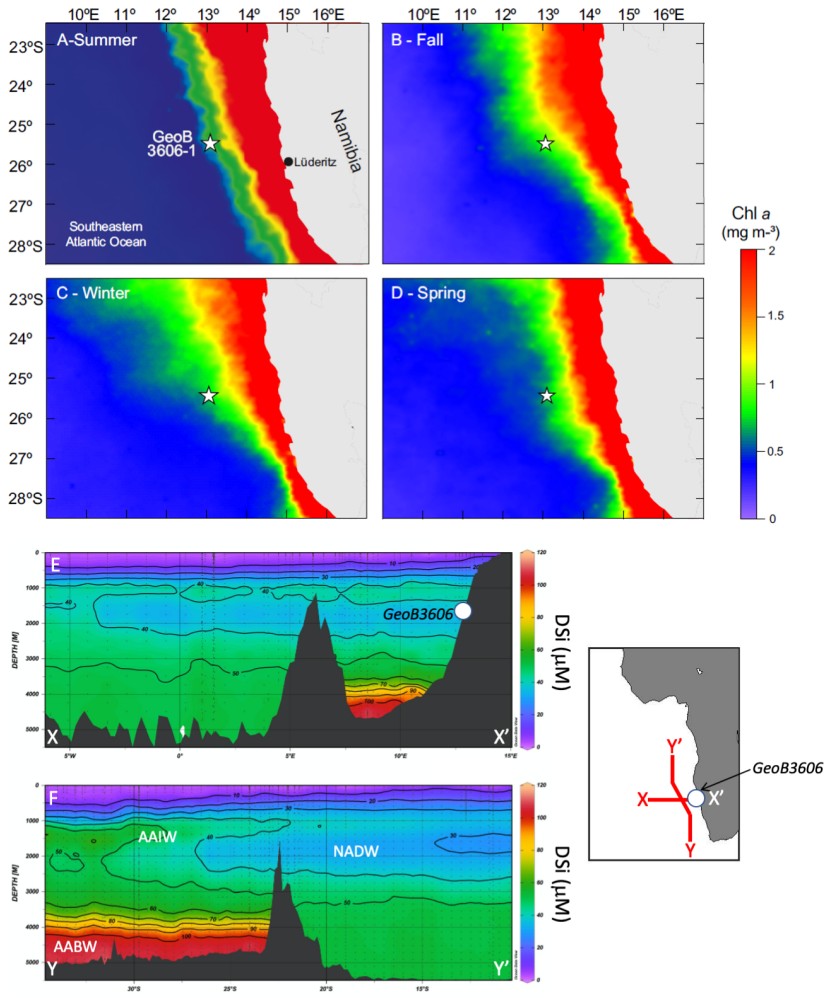

**Figure 1.** Location of the study site GeoB3606-1 (white star) in the Benguela Upwelling System. Seasonally averaged concentration of chlorophyll a (mg m$^{-3}$) for (A) January-March (austral summer), (B) April-June (austral fall), (C) July-September (austral winter), and (D) October-December (austral spring) from the years 1998-2009 in 9 by 9 km resolution (Goddard Space Flight Center, http://oceancolor.gsfc.nasa.gov/SeaWiFS). (E-F) Sections showing DSi near the core location site in the modern Atlantic. A: west-east transect (X-X'); B: south-north transect (Y-Y'). Transects shown by red lines in insert; approximate core location shown by circle.





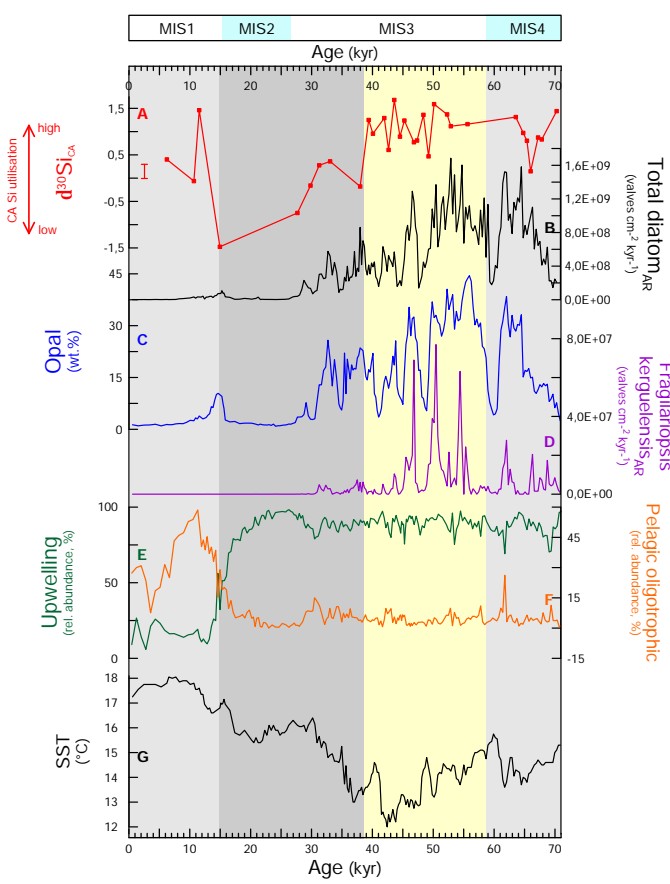

Figure 2
Hendry et al.

**Figure 2.** Comparison between diatom silicon isotope records and other archives from GeoB3606-1. A: $\delta^{30}Si_{CA}$ records from large diatoms *C. radiatus* and *A. curvatulus* (error bar shows long term reproducibility based on repeat measurements of reference standards, ± 2SD); B: total diatom accumulation rate; C: opal weight percent; D: abundance of Antarctic diatom *F. kerguelensis*; E: relative abundance of diatom species characterising coastal upwelling; F: relative abundance of diatom species characterising open ocean/pelagic conditions; G: alkenone-based sea surface temperatures (SST) (published data from MacKay et al., 2016; Romero, 2010; Romero et al., 2003, 2015). Vertical shaded bars highlight the time-periods discussed in section 3.1.





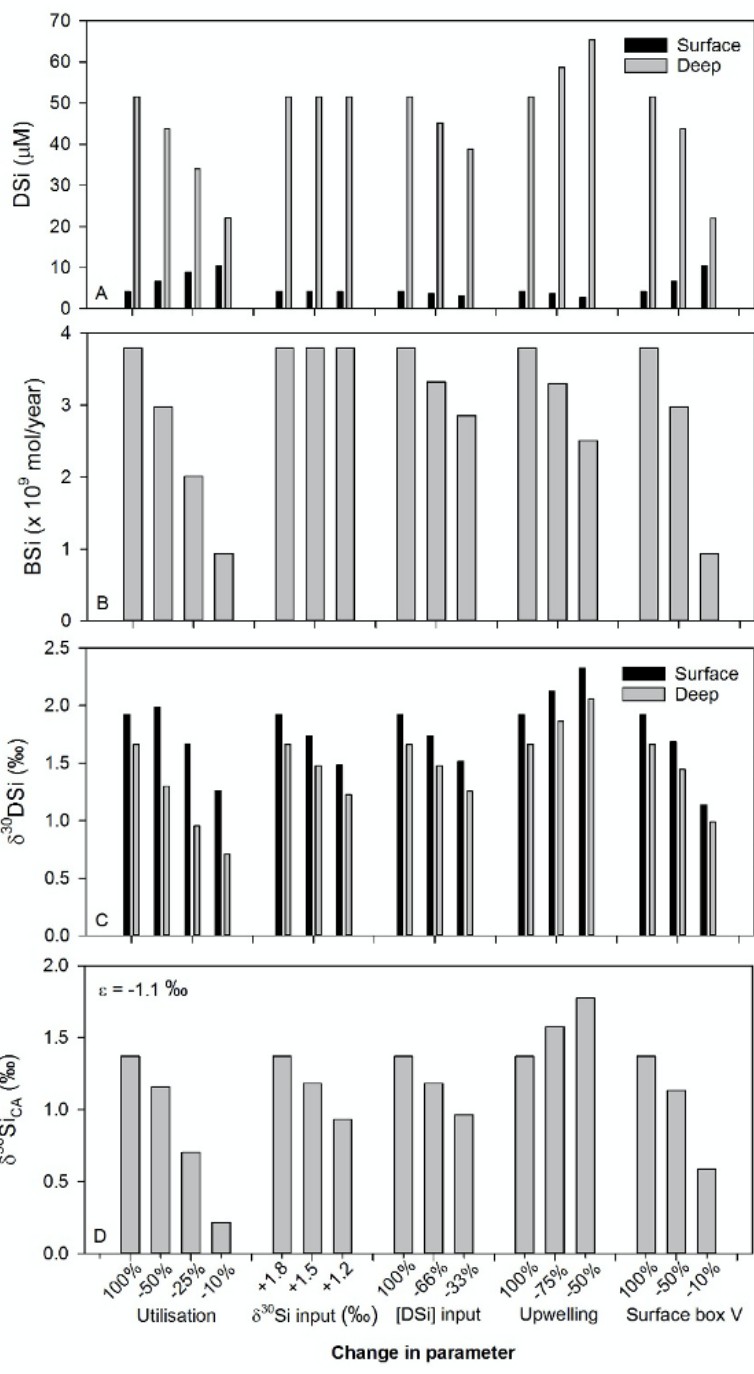

**Figure 3.** Model output results from the two-box model experiments. Each experiment shows the impact on different output parameters from changing one aspect of the model configuration (biological utilisation by dominant diatom species, isotopic and [DSi] composition of input waters reflecting a change in SO waters, upwelling strength, and the volume of the surface box): A) seawater [DSi] in the surface and deep box; B) opal production by dominant species; C) the isotopic composition of DSi in the surface and deep box; and D) the isotopic composition of the pelagic diatoms, *C. radiatus* and *A. curvatulus* ($\delta^{30}Si_{CA}$).





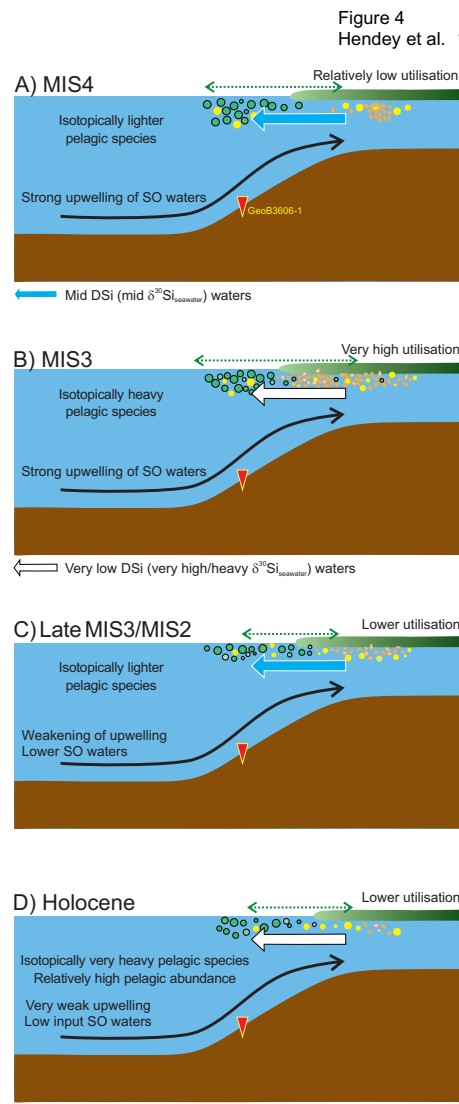

**Figure 4.** Schematics of Si cycling in the BUS at each time interval of the GeoB3606 record. A: MIS4; B: Early MIS3; C: Late MIS3/MIS2; D: MIS1 (Holocene). Full circles represent valves of *A. curvatulus* and *C. radiatus*, while small elliptic forms represent resting spores of *Chaetoceros*. Blue arrows represent the movement of DSi-rich surface waters; white arrows represent the movement of DSi-poor surface waters. See main text for discussion.





**Table 1.** Summary of Neptune Plus operating conditions.

| | |
|---|---|
| **Forward power** | 1200W |
| **Reflected power** | <2W |
| **Plasma gas** | 16 l/min |
| **Auxiliary gas flow** | 0.8 l/min |
| **Nebuliser carrier gas flow** | 1.1 l/min |
| **Nebuliser** | ESI PFA-50 (50 $\mu$l/min uptake rate) |
| **Spray chamber** | Thermo Fisher Stable Sample Introduction (SSI) quartz dual cyclonic |
| **Type of detector** | Faraday ($10^{11}\Omega$ resistors) |
| **Torch** | Demountable glass torch with quartz injector |
| **Cones** | Thermo Fisher nickel 'H' sample and skimmer |
| **Sample uptake time** | 90 seconds |
| **Wash time between samples** | 10 minutes |



**Table 2.** Summary of the acquisition method used for Si isotope analysis.

| | Low 4 cup | Low 3 cup | Axial cup | High 3 cup | High 4 cup | Integration time/(s) | Magnet Settle time /(s) |
|---|---|---|---|---|---|---|---|
| **Sequence 1** | | $^{28}$Si | $^{29}$Si | $^{30}$Si | | 16.8 | 3 |
| **Sequence 1** | $^{24}$Mg | | $^{25}$Mg | | $^{26}$Mg | 8.4 | 3 |



**Table 3.** Conditions in the box model.

The surface box is fed from upwelling waters from the deep box. Waters also escape the surface box laterally;

The deep box is fed from incoming waters from the SO and waters escape by upwelling to the surface;

Upwelling rate set at 4 x $10^{13}$ L per year;

Silicon enters the surface box from river run off (set at an estimated 1 x $10^6$ mol/year) and from upwelling;

Bulk opal BSi (by dominant upwelling species) is produced in the surface box and results in isotopic fractionation ($\varepsilon$ = -1.1 ‰);

Biological production efficiency (or utilisation) and export efficiency are adjustable. Sedimentary preservation is fixed at 3.5%;

*Coscinodiscus/Actinocyclus* consume residue (with the same relative utilisation as the other diatoms) DSi after bulk opal production, assumed to follow an open model for fractionation.