# Peer review of "Nutrient utilization and diatom productivity changes in the low-latitude SE Atlantic over the past 70 kyr: Response to Southern Ocean leakage"

_Climate of the Past, 2020_

## Referee Comment (RC1) · Anonymous Referee #1 · 6 Nov 2020

General comments: Hendry et al. presented a silicon isotopic record of near-monospecific diatoms from low-latitude SE Atlantic, to explore nutrient utilization since 70 ka. Further this information, coupled with simulation results from mass-balance experiments, provides new insights into the relation of silica cycling to upwelling intensity and silicic acid utilization. In fact, an array of publications associated with the study core series (i.e., core GeoB3606 series) has been published; thereinto several publications (e.g., Shukla and Romero, 2018) have speculated the leakage of silicic acid from Southern Ocean to low- low-latitude SE Atlantic, but it is lack of compelling evidences. Here, the authors, for the first time, provide the silicon isotopic evidence to demonstrate the influences of southern-sourced silicic acid on the diatom growth in the study area. The manuscript was well written with appropriate English usages and normal logics. The conclusions are reasonably made from the presented data, and thus I approve them. I strongly recommend the publication of this manuscript after some minor modifications suggested by me as follow.

Specific comments: 1. This manuscript contains some long sentences or long sentences with brackets, such as lines 136-138, 194-196, 219-221, and so on. Although these sentences can express what the authors want to express, their readability is weak. Therefore, I suggest the authors to rewrite them; i.e., separate one sentence to more. 2. For the study core, there are enough relevant publications to provide the background on paleoenvironmental and palaeoceanographic conditions. The authors always directly cite this background information without the details. For example, the author used the SST as a proxy of upwelling intensity, but they did not explain why the SST can reflect the upwelling conditions in the study area? Not all the readers are familiar with the study area and the study core. Thus, I advise the authors to give some details when citing some important conclusive information from other publications to support their discussion. 3. To confirm the leakage of silicic acid from the Southern Ocean, the author combined the information from nutrient utilization, diatom assemblage, upwelling intensity, and so on. It is right! Other way is focus on isotopic tracing. Detailedly the author can also try to compare the silicon isotope ($\delta$30Si) values among the diatoms (A. curvatulus+C. radiatus), the waters in the study area, and the southern-sourced waters, based on the silicon isotope fractionation and water mixing. I strongly recommend the authors to have a shot, but I do not guarantee its success.

Technical corrections: Line 9: It is not appropriate to state '. . .species-specific $\delta$30Si. . .' because the two species A. curvatulus and C. radiatus were used to analyze the $\delta$30Si. Line 32: In '. . .atmospheric pCO2. . .', the 'p' should be italic. Lines 71-72: How the SST changes can account for the diatom productivity? Lines 73-75: Please explain

this sentence with some details. Line 163: Please add '.' in the end of this sentence. Line 170: What is the meaning of '. . .of what was available'? Please rewrite it. Lines 213-214: Please add the references for '. . .consistent with leakage of SO waters at this time into the eastern basin of the South Atlantic'.

---

## Referee Comment (RC2) · Anonymous Referee #2 · 25 Dec 2020

The authors present new Si isotope data from core GeoB3606-1 from the Benguela Upwelling System covering the past ∼70 kyr, since MIS 3-4, which is influenced by variations in water supply/leakage of nutrient-rich waters from the Southern Ocean via Antarctic Intermediate Water. This is generally a very interesting paper. As nicely outlined in the manuscript, leakage of nutrient-rich waters from the Southern Ocean has been studied and discussed widely in the past in order to explain variations in $CO_2$-drawdown during cold climate periods. However global records between ocean basins and regions vary widely, some indicating addition of nutrients and stimulation of primary

production in the lower latitudes, whereas others do not. Eastern boundary upwelling systems, and here especially the Benguela upwelling system, are ideal regions to study leakage theories, because it receives direct water supply from the Southern Ocean via shoaling/upwelling of nutrient-rich Antarctic water. This nutrient rich water is source for intense primary production. And diatoms in upwelling regions have been shown before to react very sensitive to changes in nutrient supply, whereby stable Si isotopes can be used to reconstruct relative utilization of the available nutrient pool. Therefore, the study presented here is of very wide interest and can potentially provide highly useful information for paleoceanographers and paleoclimate studies providing important information on the response of the Benguela upwelling region to nutrient leakage from the Southern Ocean and its effect on the global climate system. Specifically, the paper presents Si isotope data from two large diatom species, which were hand-picked from the sediment samples in order to prevent influence of changing diatom assemblages due to variable environmental conditions and potential species-specific fractionation effects. Furthermore, the Si isotope data are interpreted in context of existing records for bSi content/accumulation, sea surface temperatures (upwelling intensity) etc. Although I generally appreciate the study and the applied methods very much, I have two major concerns about details of the method and interpretation of the Si isotope data that need to be addressed before publication in Climate of the Past.

Major Comments: The authors present a "species-specific" diatom Si isotope record, consisting of two diatom species (A. curvatulus and C. radiatus), which, as far as I understand, grow under and therefore represent different environmental conditions concerning nutrient status, temperature preference, etc. The Si isotope data are, however, presented as a single isotope record. The diatoms were hand-picked from the record. I generally think is a very good approach for the area and the research question, but it is completely unclear to me how the authors "mixed" the specimens in each sample, i.e. did they made sure to always have the same exact number of specimens from each of the two species in each sample? Could there have been an imbalance according to variations in the relative number of specimens or variations in their size/silicification

in each sample over time? The authors acknowledge themselves that we have indications for species-specific fractionation effects for Si isotope in diatoms. Also, the fractionation behaviour for diatoms might change significantly under variable environmental conditions (high/low nutrient concentrations in the surrounding seawater with effects on the influx:efflux ratio of dissolved Si for single diatom cells, potentially affecting the preserved Si isotope ratios).

The total variation in the d30Si record is 3 per mil, ranging from -1.5 to +1.5. This range is huge compared to any other diatom record published so far. Admittedly, single-species record can have larger variations compared to mixed-species, where variations are rather flattened out. However, to me the explanation/interpretation of this range, and especially the negative values, is quite superficial. The authors assume a constant diatom Si isotope fractionation of -1.1 per mil, and also acknowledge larger possible species-specific fractionation. That's okay. However, assuming the -1.1 per mil fractionation, the most negative values in the record would imply surface water d30Si values of -0.4 per mil. This is completely unrealistic. Even in combination with a larger isotope fractionation between diatoms and seawater and other environmental effects in the upwelling region (only mentioned very broadly), I don't see how this is possible. Please explain.

Additional minor comments: L. 116: delete "(multi-collector-inductively coupled plasma mass spectrometer)", abbreviation has been introduced before L. 198: "Quantifying changes to the . . ." I'm not sure that I see the "quantification" in this section. L. 200: is river input a significant source for dSi in the BUS? Also, what about dust from the arid hinterland? Dust storms towards the upwelling region/open ocean are a regular seasonal feature in the region at least nowadays. Any proposed changes over time there? Could there be variations in Si utilization due to changes in Fe fertilization from dust directly in the region, not just leakage?

---

## Author Comment (AC1) · 8 Jan 2021

**1   Response to Reviewer 1**

Hendry et al. presented a silicon isotopic record of near- monospecific diatoms from low-latitude SE Atlantic, to explore nutrient utilization since 70 ka. Further this information, coupled with simulation results from mass-balance experiments, provides new insights into the relation of silica cycling to upwelling intensity and silicic acid

utilization. In fact, an array of publications associated with the study core series (i.e., core GeoB3606 series) has been published; there into several publications (e.g., Shukla and Romero, 2018) have speculated the leakage of silicic acid from Southern Ocean to low- low-latitude SE Atlantic, but it is lack of compelling evidences. Here, the authors, for the first time, provide the silicon isotopic evidence to demonstrate the influences of southern-sourced silicic acid on the diatom growth in the study area. The manuscript was well written with appropriate English usages and normal logics. The conclusions are reasonably made from the presented data, and thus I approve them. I strongly recommend the publication of this manuscript after some minor modifications suggested by me as follow.

**We would like to thank the reviewer for their positive comments and will address the minor modifications below.**

Specific comments: 1. This manuscript contains some long sentences or long sentences with brackets, such as lines 136-138, 194-196, 219-221, and so on. Although these sentences can express what the authors want to express, their readability is weak. Therefore, I suggest the authors to rewrite them; i.e., separate one sentence to more.

**Many thanks for the constructive criticism. We have shortened the sentences as suggested, and have checked throughout the manuscript for readability and typos (changes made on lines 32-33, line 54, and line 220).**

2. For the study core, there are enough relevant publications to provide the background on paleoenvironmental and palaeoceanographic conditions. The authors always directly cite this background information without the details. For example, the

author used the SST as a proxy of upwelling intensity, but they did not explain why the SST can reflect the upwelling conditions in the study area? Not all the readers are familiar with the study area and the study core. Thus, I advise the authors to give some details when citing some important conclusive information from other publications to support their discussion.

**We have addressed this concern by adding in further explanation to the proxies as requested. We have not included a detailed explanation of how the Uk-37 alkenone proxy records SST as it is an established proxy that has been used in the region before. The section now reads:**
*"Changes in nutrient supply as a result of enhanced mixing of the upper-most water column, indicated by reduced SST reconstructions from alkenone archives, are unlikely to fully explain variations in diatom productivity alone*...
***The inverse correlation between the relative abundance of the Antarctic diatom Fragilariopsis kerguelensis and the alkenone-based SST variations (Fig. 2) in GeoB3606-1 from 70 to 30 kyr suggests a combination of enhanced DSi-rich SO water invasion and stronger wind-driven mixing respectively during this interval of high opal burial*...*"**

3. To confirm the leakage of silicic acid from the Southern Ocean, the author combined the information from nutrient utilization, diatom assemblage, upwelling intensity, and so on. It is right! Other way is focus on isotopic tracing. Detailedly the author can also try to compare the silicon isotope ($\delta^{30}Si$) values among the diatoms (A. curvatulus+C. radiatus), the waters in the study area, and the southern-sourced waters, based on the silicon isotope fractionation and water mixing. I strongly recommend the authors to have a shot, but I do not guarantee its success.

**We agree that it is useful to compare the silicon concentrations and isotopic**

**compositions of the diatoms, with likely compositions of the seawater and end-members. We carried out modelling "thought-experiments" to investigate the potential interpretations of the downcore data. The motivation behind the modelling study is now emphasised on line 226 onwards (see also response to reviewer 2).**

Technical corrections: Line 9: It is not appropriate to state '. . .species-specific $\delta$30Si. . .' because the two species A. curvatulus and C. radiatus were used to analyze the $\delta$30Si.

**This has been corrected, and the sentence now reads:**
*"Here, we present the first combined $\delta^{30}Si$ record of two large centric diatoms from the BUS, encompassing full glacial conditions to the Holocene."*

Line 32: In '...atmospheric pCO2...', the 'p' should be italic.

**This has been corrected.**

Lines 71-72: How the SST changes can account for the diatom productivity?

**This has been clarified, see above.**

Lines 73-75: Please explain this sentence with some details.

**This has been clarified, see above.**

Line 163: Please add ':' in the end of this sentence.

**This has been corrected.**

Line 170: What is the meaning of '. . .of what was available'? Please rewrite it.

**This has been clarified. The sentence now reads:**
*"Not only was Chaetoceros production high because of the rate of supply of DSi to coastal waters, but also because they were able to use a high proportion of this available DSi."*

Lines 213-214: Please add the references for '. . .consistent with leakage of SO waters at this time into the eastern basin of the South Atlantic'.

**Many thanks for this comment. We were unclear that the statement was relating to interpretation of our own data rather than referring to an existing study. We have clarified this, and the sentence now reads:**
*"We deduce from the $\delta^{30}Si_{CA}$ record that there was strong but variable upwelling of Si-rich waters during MIS4 and MIS3, consistent with an interpretation that SO water leaked into the eastern basin of the South Atlantic at this time."*

---

## Author Comment (AC2) · 8 Jan 2021

**1 Response to Reviewer 2**

The authors present new Si isotope data from core GeoB3606-1 from the Benguela Upwelling System covering the past âLij70 kyr, since MIS 3-4, which is influenced by variations in water supply/leakage of nutrient-rich waters from the Southern Ocean via Antarctic Intermediate Water. This is generally a very interesting paper. As nicely

outlined in the manuscript, leakage of nutrient-rich waters from the Southern Ocean has been studied and discussed widely in the past in order to explain variations in CO2- drawdown during cold climate periods. However global records between ocean basins and regions vary widely, some indicating addition of nutrients and stimulation of primary production in the lower latitudes, whereas others do not. Eastern boundary upwelling systems, and here especially the Benguela upwelling system, are ideal regions to study leakage theories, because it receives direct water supply from the Southern Ocean via shoaling/upwelling of nutrient-rich Antarctic water. This nutrient rich water is source for intense primary production. And diatoms in upwelling regions have been shown before to react very sensitive to changes in nutrient supply, whereby stable Si isotopes can be used to reconstruct relative utilization of the available nutrient pool. Therefore, the study presented here is of very wide interest and can potentially provide highly useful information for paleoceanographers and paleoclimate studies providing important in- formation on the response of the Benguela upwelling region to nutrient leakage from the Southern Ocean and its effect on the global climate system. Specifically, the paper presents Si isotope data from two large diatom species, which were hand-picked from the sediment samples in order to prevent influence of changing diatom assemblages due to variable environmental conditions and potential species-specific fractionation effects. Furthermore, the Si isotope data are interpreted in context of existing records for bSi content/accumulation, sea surface temperatures (upwelling intensity) etc. Although I generally appreciate the study and the applied methods very much, I have two major concerns about details of the method and interpretation of the Si isotope data that need to be addressed before publication in Climate of the Past.

We would like to thank the reviewer for their positive comments and will address their recommendations for modifications below.
Major Comments: The authors present a "species-specific" diatom Si isotope record, consisting of two diatom species (*A. curvatulus and C. radiatus*), which, as far as I understand, grow under and therefore represent different environmental conditions concerning nutrient status, temperature preference, etc. The Si isotope data are, however, presented as a single isotope record. The diatoms were hand-picked from the record. I generally think is a very good approach for the area and the research question, but it is completely unclear to me how the authors "mixed" the specimens in each sample, i.e. did they made sure to always have the same exact number of specimens from each of the two species in each sample? Could there have been an imbalance according to variations in the relative number of specimens or variations in their size/silicification in each sample over time? The authors acknowledge themselves that we have indications for species-specific fractionation effects for Si isotope in diatoms.

We thank the reviewer for these valid comments. Whilst it is not ideal to mix two species we justify the approach here because i) we needed to obtain sufficient material for analysis and ii) these species are both large centric diatoms with similar ecologies. In the region of the BUS, the major ecological divide for phytoplankton relates to whether or not the species in question resides inside or outside of the main upwelling zone. Both of the diatom species in question here are common in more pelagic waters outside of the upwelling zone.

This has now been emphasised on line 90:

"Compared to A. curvatulus, it represents a more "pelagial" signal. However, both species can be considered as occupying niches outside of the main upwelling zone"

We have also removed reference to the term 'species-specific' where this could be interpretated as referring to our records (abstract and line 273). We have also emphasised on line 104:

"Valves of the two diatom species were combined to allow for enough ma-

CPD
terial and because the species diatoms are indistinguishable under a lowmagnification binocular microscope (note that low MIS2 data resolution was due to limited valve availability). They were the two only large centric diatoms present in the washed/sieved samples of GeoB3606-1."

Also, the fractionation behaviour for diatoms might change significantly under variable environmental conditions (high/low nutrient concentrations in the surrounding seawater with effects on the influx:efflux ratio of dissolved Si for single diatom cells, potentially affecting the preserved Si isotope ratios). We agree entirely, and we explore these potential changes in environmental conditions in the modelling thought-experiments. The latter suggestion of investigating the impact of influx:efflux on silicon isotope ratios is very poorly constrained in biological systems and we feel this is out of the scope of this study.

The total variation in the d30Si record is 3 per mil, ranging from -1.5 to +1.5. This range is huge compared to any other diatom record published so far. Admittedly, single-species record can have larger variations compared to mixed-species, where variations are rather flattened out. However, to me the explanation/interpretation of this range, and especially the negative values, is quite superficial. The authors assume a constant diatom Si isotope fractionation of -1.1 per mil, and also acknowledge larger possible species-specific fractionation. That's okay. However, assuming the -1.1 per mil fractionation, the most negative values in the record would imply surface water  $\delta^{30}Si$  values of -0.4 per mil. This is completely unrealistic. Even in combination with a larger isotope fractionation between diatoms and seawater and other environmental effects in the upwelling region (only mentioned very broadly), I don't see how this is possible. Please explain.

We agree that the record shows very large isotopic variability. As there is no straightforward explanation, we decided to explore possible explanations

**CPD**
with the modelling thought-experiments. We conclude that it's only possible to explain the very light isotope ratios through a combination of changes in upwelling and a divergence fractionation factor (see line 236).

We have emphasised the motivation behind the modelling on line 226:

"Our downcore record reveals strong variability in silicon isotope systematics in the BUS over the late Quaternary, which is challenging to interpret by simple changes one of the many different potential environmental driving mechanisms (e.g. upwelling intensity, biological productivity, oceanic end-member compositions)"

And discussed the challenges with interpreting the light isotopic signature (and, so, the range in isotopic values) on line 235:

"A combination of changes in upwelling intensity, stratification, seawater input and utilisation can act together to change the isotopic differentiation of shallow and deep-water masses. The experiment results also indicate that, in order to achieve the extremely low  $\delta^{30}Si_{CA}$  values observed downcore, isotopic fractionation during DSi uptake by A. curvatulus and C. radiatus is likely to be greater than generally assumed, up to -2, as observed in some diatom cultures of other species..."

Additional minor comments: L. 116: delete "(multi-collector-inductively coupled plasma mass spectrometer)", abbreviation has been introduced before

Done.

L. 198: "Quantifying changes to the . . ." I'm not sure that I see the "quantification" in this section.
**This has been changed to "Exploring changes to the..." (now in line 225).**

L. 200: is river input a significant source for dSi in the BUS? Also, what about dust from the arid hinterland? Dust storms towards the upwelling region/open ocean are a regular seasonal feature in the region at least nowadays. Any proposed changes over time there? Could there be variations in Si utilization due to changes in Fe fertilization from dust directly in the region, not just leakage?

This is a good point, which we are happy to discuss and clarify. There is no evidence for a significant fluviatile input along the Namibian coast during the late Quaternary (Shi et al., 2001). Regarding winds, sedimentological studies conducted on the upper Namibian slope (core drilled around 1,000 m water depth, shallower than GeoB3606-1; Pichevin et al., 2005) shows a strong match between stronger wind/drier land conditions and the overall trend of highest diatom values at site GeoB3606-1 from 70 kyr to 36 kyr is evidence for a trade wind effect on the diatom production. Similarly, the increase of SST at site GeoB3606-1 (Romero et al., 2015) corresponds well with the weakened trades intensity after 36 kyr. Although the strength of the trade winds remained strong during MIS2 (Shi et al., 2001), upwelling conditions in surface waters overlying the lower slope off SW Africa became less favourable for diatom production. In other words, the diatoms are responding to upwelling and DSi input more than the trade winds (i.e. the dust supply is not sufficient to promote diatom production).

We have added to the methods section on line 145:

"There is no evidence for a significant fluviatile input along the Namibian coast
during the late Quaternary (Shi et al., 2001). As such, relatively low terrestrial DSi inputs were set as a constant in the model."

We have added the following on line 179:

"The high DSi supply could also have been promoted by strong trade winds, due to enhanced dust supply as a result of drier conditions on land in addition to upwelling of marine sources (Shi et al., 2001; Pichevin et al., 2005)."

And on line 205:

"Although upwelling conditions in surface waters overlying the lower slope off SW Africa became less favourable for diatom production, the strength of the trade winds remained strong during MIS2 (Shi et al., 2001), indicating that dust supply is a secondary control on diatom activity in the Late Quaternary in the SE Atlantic."

Additional references:

Pichevin, L., Cremer, M., Giraudeau, J., Bertrand, P., 2005b. A 190 kr record of lithogenic grain-size on the Namibian slope: Forging a tigh link between past wind-strength and coastal upwelling dynamics. Marine Geology 218, 81-96.

Shi, N., Schneider, R.R., Bueg, H.-J., Dupont, L.M., 2001. Southeast trade wind variations during the last 135 kyr: evidence from pollen spectra in eastern South Atlantic sediments. Earth Planet. Sci. Lett. 187, 311-321.

---

## Author Response (AR1)

**Nutrient utilization and diatom productivity changes in the low-latitude SE Atlantic over the past 70 kyr: Response to Southern Ocean leakage**

Katharine Hendry[1], Oscar Romero[2,3], and Vanessa Pashley[4]

[1]University of Bristol, School of Earth Sciences, Wills Memorial Building, Queen's Road, Bristol, BS8 1RJ, UK
[2]MARUM–Center for Marine Environmental Sciences, Leobener Str. 8, University of Bremen, 28359 Bremen, Germany
[3]Alfred Wegener Institute, Helmholtz Centre for Polar and Marine Research, 27568 Bremerhaven, Germany
[4]Geochronology and Tracers Facility, British Geological Survey, Keyworth, NG12 5GG, UK

**Correspondence:** Katharine Hendry (K.Hendry@bristol.ac.uk)

**1 Response to Editor**

Two reviewers provided constructive comments and recommendations for the publication. The reviewers gave positive comments about the value of the data and the interesting interpretations. Queries around use of 2 specific for isotope work have been addressed in the author replies, including clarifying that the isotope data is not 'species-specific'. Further concerns about citing contextual information for the area has been addressed, as well as those related to the processes being explored with the simple model.

**Many thanks to the editor for the thorough review of our article. We are happy to respond to the further comments below. We have uploaded a new version of the manuscript with changes to address the comments from the two reviewers and the editor highlighted in red, blue and cyan respectively.**

1) Links between what is happening in the Benguela upwelling system and in the Southern Ocean, the source of the DSi. At present the data and it's interpretation are presented without reference to knowledge of any changes in SO circulation / productivity, which may offer support or explanation to the patterns the authors propose. It may be that such data does not exist (e.g. at the resolution the authors work here) but the text in section 3.1 is very focussed only on the Benguela system, e.g. lines 174-175 notes "a decrease in the DSi concentration of the supplied water...": is there data from the Southern Ocean marine archives which could predict/explain why this would happen or is it a signal driven by the local (upwelling) circulation changing? Do other Benguela sites offer support/explanation?
line 183-184: as for my previous comment, are there other data sets which support or explain why this supply changed?

**This is an important limitation, and this issue is discussed in our previous publications (Romero et al., 2003; Romero, 2010; Romero et al., 2015; Shukla and Romero, 2018). However, our understanding of SO nutrient dynamics over these**

timescales has so far limited by the lack of published, similarly high-resolved record of diatom production south of the Polar Front for the past 70 kyr.

25

We have made this point on line 39 onwards:

*"However, as yet there are no published archives of silicon cycling under full glacial conditions south of the subantartic front. Furthermore, very few archives have been published over this time period from lower latitudes..."*

30

Instead, we have to rely on information gleaned from core sites in the lower latitudes.

We postulate that intermittent, SO-originated pulses of DSi into the BUS between 70 kyr and ca. 30 kyr led to the upwelling of Si-rich waters off Namibia (Romero et al., 2003; Romero, 2010; Romero et al., 2015). This nutrient sce-
35 nario was triggered by the equatorward transport of Si-enriched waters of Antarctic origin, either by (i) direct mixing or by (ii) the advection of Subantarctic Mode Waters (whose present-day Si content is low relative to surrounding water masses; Matsumoto et al., 2002) that invaded the middle to lower thermocline of subtropical coastal upwelling areas (Sarmiento et al., 2004). The equatorward leakage of DSi followed intervals of lowered diatom productivity in the SO south of the Subantarctic Front due of varying physical and biological conditions (sea ice cover, winds, Fe input; Mat-
40 sumoto et al., 2014). Two possible drawbacks of this sub-Milankovitch scale leakage scenario are (i) the lack of a diatom reconstruction south of the Subantarctic Front showing millennial-scale variability, and (ii) the prediction of glacial increases and interglacial decreases of Si leakage. Additional evidence for a non-glacial, sub-Milankovitch Si leakage is provided by increased opal burial recorded in the eastern equatorial Pacific between 40-60 kyr, attributed to extended sea ice around Antarctica (as already discussed in the introduction).
45 Another possible link between SO and the BUS is provided by the weak anti-correlation between the size and concentra-tion of *F. kerguelensis'* valves at GeoB3606-1 during the late MIS3 (Shukla and Romero, 2018). This could have resulted from the higher Fe availability through dust flux during 70–30 kyr in the SO (i.e., Mahowald et al., 2005; Martínez-García et al., 2014), which might have caused increased growth rates of diatoms and a decrease in the silicate-to-nitrate uptake ratio of diatoms (Matsumoto et al., 2002; Brzezinski et al., 2002; Matsumoto et al., 2014). The magnitude of
50 any silicic acid leakage is largely dependent on the behavior of AAIW and SAMW (Crosta et al., 2007) which is largely unknown for glacial-interglacial period (Kohfeld et al., 2013). Although we are not able to provide a convincing expla-nation of the intensity changes of AAIW or SAMW in our *F. kerguelensis* valve size data (which was anyway beyond the scope of that work), both the valve size and concentration *F. kerguelensis* in core GeoB3606-1 suggest decreased silicate:nitrate uptake of diatoms in presence of higher iron availability through dust flux (Takeda, 1998; Hutchins and
55 Bruland, 1998).

**We have added the following to line 207 onwards:**

*"A weak anti-correlation between the size and concentration of F. kerguelensis valves at GeoB3606-1 during the late MIS3 (Shukla and Romero, 2018) could indicate an increase in growth rates of these diatoms resulting from a higher Fe availability in the SO due to an enhanced supply of dust (Martínez-García 210 et al., 2014). Alleviation of Fe limitation in the SO could also have caused a decrease in the DSi-to-nitrate uptake ratio of diatoms and, so, a relative enrichment of DSi in SO waters exported to the lower latitudes (Brzezinski et al., 2002). Although upwelling conditions in surface waters overlying the lower slope off SW Africa became less favourable for diatom production, the strength of the trade winds remained strong during MIS2 (Shi et al., 2001), indicating that lower latitude dust supply is a secondary control on diatom activity in the Late Quaternary in the SE Atlantic."*

**We have expanded the discussion of the late MIS2 to include more information about SO records on line 214 onwards:**

*"The decreased delivery of DSi into the SE Atlantic around 17 kyr led to the floral shift at GeoB3606-1 (Fig. 2E). Higher CaCO3 (lower opal) values at GeoB3606-1 from late MIS2 to the mid/late Holocene (Romero et al., 2015) indicate a shift in predominant nutrients toward Si-depleted waters. Following the lessened sea ice cover in the SO (Crosta et al., 2005), and the lowered input of Fe south of the Polar Front due to weakened wind intensity during the last deglaciation (Kohfeld et al., 2005; Sijp and England, 2008), DSi was mainly consumed in waters south of the Subantarctic Front and became mostly trapped in underlying sediments (Brzezinski et al., 2002; Bradtmiller et al., 2009). This scenario corresponds to the present-day dynamics of production and sedimentation of biogenic particulates in the southern BUS (Romero and Armand, 2010), where coccolithophorids dominate primary production over diatoms."*

2) lines 66-67: since there has been a revision to the radiocalibration curves since the McKay et al. 2016 publication could the authors please clarify here which calibration model was used to generate the calendar ages? (rather than the reader having to seek out the original age model papers)

**The age model we used here is that of McKay et al., 2016, and the calibration process is now detailed on line 69:**

*"The radiocarbon ages of all samples were converted into calendar years, and a new age model (McKay et al., 2016) was created using the OXCAL 4.2 program with the marine calibration curve MARINE13 (Reimer et al., 2013)."*

3) response to R1 regarding SST and upwelling: "reduced SST reconstructions" could be read as "fewer SST data points". "...reduced SSTs as reconstructed from alkenone archives" might be more clear? The authors may also consider line 160-161: "Cooler SSTs indicate a pulse of upwelling..."?

**Many thanks for these suggestions. We have changed these lines as requested.**

4) response to R2 regarding Fe fertilisation: the authors note an addition on Line 205 to address this concern. Although this addition refers to 'dust supply' it does not indicate whether this refers to the concern about Fe fertilisation which was expressed by the reviewer. Could the authors consider clarifying here if they are arguing for a lack of Fe fertilisation when they refer to dust, or are they referring to DSi and/or Fe supply?

**Dust supplies to surface waters have been suggested to alleviate both Si and Fe limitation. We have clarified that we were referring to both possibilities on line 181**

*"In addition to upwelling of marine sources, strong trade winds could also have promoted diatom productivity through the supply of DSi and trace nutrients to surface waters via dust, given the drier conditions on land..."*

**Additional references added to the manuscript:**

**Bradtmiller, L.I., Anderson, R.F., Fleisher, M.Q., Burckle, L.H. 2009. Comparing glacial and Holocene opal fluxes in the Pacific sector of the Southern Ocean. Paleoceanography 24, PA2214, doi:10.1029/2008PA001693.**

**Crosta, X., Shemesh, A., Etourneau, J., Yam, R., Billy, I., Pichon, J.J., 2005. Nutrient cycling in the Indian sector of the Southern Ocean over the last 50,000 years. Global Biogeochemical Cycles 19, doi:10.1029/2004GB002344.**

**Kohfeld, K.E., Le Quéré, C., Harrison, S.P., Anderson, R.F., 2005. Role of Marine Biology in Glacial-Interglacial CO2 Cycles. Science 308, 74-78.**

**Martínez-García, A., Sigman, D.M., Ren, H., Anderson, R.F., Straub, M., Hodell, D.A., Jaccard, S.L., Eglinton, T.I., Haug, G.H., 2014. Iron fertilization of the Subantarctic Ocean during the last ice age. Science 343, 1347–1350. http://dx.doi.org/10.11 science.1246848.**

**Sijp, W., England, M.H., 2008. The effect of a northward shift in the southern hemisphere westerlies on the global ocean. Progress in Oceanography 79, 1–19.**

**Reimer, P. J., et al. (2013), IntCal13 and Marine13 radiocarbon age calibration curves 0–50,000 years cal BP, Radiocarbon, 55(4), 1869–1887**

**Other references cited in this response:**

**Brzezinski, M.A., Pride, C.J., Sigman, D.M., Sarmiento, J.L., Matsumoto, K., Gruber, N., Rau, G.H., Coale, K.H., 2002. A switch from Si(OH)4 to NO3 depletion in the glacial Southern Ocean. Geophysical Research Letters 29, doi:1510.1029/2001GL014349.**

**Hutchins, D.A., Bruland, K.W., 1998. Iron-limited diatom growth and Si:N uptake ratios in a coastal upwelling regime.**

Nature 393, 561–564.

Kohfeld, K., Graham, R.M., de Boer, A.M., Sime, L., Wolff, E., Le Quere, C., 2013. Glacialinterglacial changes in southern hemisphere westerly winds: paleo-data synthesis.

Mahowald, N.M., Baker, A.R., Bergametti, G., Brooks, N., Duce, R.A., Jickells, T.D., Kubilay, N., Prospero, J.M., Tegen, T., 2005. Atmospheric global dust cycle and iron inputs to the ocean. Glob. Biogeochem. Cycles 19. http://dx.doi.org/10.1029/2004GB002402.

Matsumoto, K., Sarmiento, J.L., Brzezinski, M.A., 2002. Silicic acid leakage from the Southern Ocean: A possible explanation for glacial atmospheric pCO2. Global Biogeochem. Cycles 16, doi:10.1029/2001GB001442.

Matsumoto, K., Chase, Z., Kohfeld, K., 2014. Different mechanisms of silicic acid leakage and their biogeochemical consequences. Paleoceanography 20, 238–254, doi:210.1002/2013PA002588.

Romero, O.E., Armand, L.K., 2010. Marine diatoms as indicators of modern changes in oceanographic conditions. In: Smol, J.P., Stoermer, E.F., (Eds.), The diatoms: Applications for the Environmental and Earth Sciences (2nd Ed.). Cambridge University Press, U.K.

Sarmiento, J.L., Gruber, N., Brzezinski, M.A., Dunne, J.P., 2004. High-latitude controls of thermocline nutrients and low latitude biological productivity. Nature 427, 56–60.

Takeda, S., 1998. Influence of iron availability on nutrient consumption ratio of diatoms in oceanic waters. Nature 393, 774–777.